# Liraglutide Treatment Restores Cardiac Function After Isoprenaline-Induced Myocardial Injury and Prevents Heart Failure in Rats

**DOI:** 10.3390/life15030443

**Published:** 2025-03-12

**Authors:** Zorislava Bajic, Tanja Sobot, Aleksandra Smitran, Snezana Uletilovic, Nebojša Mandić-Kovačević, Tanja Cvjetkovic, Ugljesa Malicevic, Bojan Stanetic, Đorđe Đukanović, Milka Maticic, Sanja Jovicic, Dragan M. Djuric, Milos P. Stojiljkovic, Ranko Skrbic

**Affiliations:** 1Department of Physiology, Faculty of Medicine, University of Banja Luka, 78000 Banja Luka, The Republic of Srpska, Bosnia and Herzegovina; tanja.sobot@med.unibl.org; 2Centre for Biomedical Research, Faculty of Medicine, University of Banja Luka, 78000 Banja Luka, The Republic of Srpska, Bosnia and Herzegovina; snezana.uletilovic@med.unibl.org (S.U.); nebojsa.mandic-kovacevic@med.unibl.org (N.M.-K.); tanja.cvjetkovic@med.unibl.org (T.C.); ugljesa.malicevic@med.unibl.org (U.M.); djordje.djukanovic@med.unibl.org (Đ.Đ.); milka.maticic@med.unibl.org (M.M.); sanja.jovicic@med.unibl.org (S.J.); milos.stojiljkovic@med.unibl.org (M.P.S.); ranko.skrbic@med.unibl.org (R.S.); 3Department of Microbiology and Immunology, Faculty of Medicine, University of Banja Luka, 78000 Banja Luka, The Republic of Srpska, Bosnia and Herzegovina; aleksandra.smitran@med.unibl.org; 4Department of Medical Biochemistry and Chemistry, Faculty of Medicine, University of Banja Luka, 78000 Banja Luka, The Republic of Srpska, Bosnia and Herzegovina; 5Department of Pharmacy, Faculty of Medicine, University of Banja Luka, 78000 Banja Luka, The Republic of Srpska, Bosnia and Herzegovina; 6Department of Pathophysiology, Faculty of Medicine, University of Banja Luka, 78000 Banja Luka, The Republic of Srpska, Bosnia and Herzegovina; 7Department of Cardiology, University Clinical Centre of the Republic of Srpska, 78000 Banja Luka, The Republic of Srpska, Bosnia and Herzegovina; bojan.stanetic@kc-bl.com; 8Department of Pharmaceutical Chemistry, Faculty of Medicine, University of Banja Luka, 78000 Banja Luka, The Republic of Srpska, Bosnia and Herzegovina; 9Department of Histology and Embryology, Faculty of Medicine, University of Banja Luka, 78000 Banja Luka, The Republic of Srpska, Bosnia and Herzegovina; 10Faculty of Medicine, Institute of Medical Physiology “Richard Burian”, University of Belgrade, 11000 Belgrade, Serbia; dr_djuric@yahoo.com; 11Department of Pharmacology, Toxicology and Clinical Pharmacology, Faculty of Medicine, University of Banja Luka, 78000 Banja Luka, The Republic of Srpska, Bosnia and Herzegovina; 12Academy of Science and Arts of the Republic of Srpska, 78000 Banja Luka, The Republic of Srpska, Bosnia and Herzegovina; 13Department of Pathologic Physiology, I.M. Sechenov First Moscow State Medical University, 119435 Moscow, Russia

**Keywords:** isoprenaline-induced heart failure, liraglutide, oxidative stress, electrocardiography, echocardiography

## Abstract

Background: Myocardial injury (MI) is characterized by an increased level of at least one cardiac troponin. Experimental MI can be induced by isoprenaline, a β-adrenergic agonist, and it can lead to heart failure (HF). Liraglutide is glucagon-like 1 peptide receptor agonist used in diabetes management, but it has anti-inflammatory and antioxidative effects, which can be beneficial in treatment of HF. The aim of this study was to investigate the effects of liraglutide on isoprenaline-induced MI and prevention of HF. Methods: Male Wistar albino rats were divided into four groups: Con—received saline the first 2 days + saline the next 7 days; Iso—isoprenaline the first 2 days + saline the next 7 days; Lir—saline the first 2 days + liraglutide the next 7 days; Iso + Lir—isoprenaline the first 2 days + liraglutide the next 7 days. On day 10, blood samples were taken for biochemical analysis and oxidative stress marker evaluation, and hearts were isolated for pathohistological analysis. Cardiac function was assessed by electrocardiography (ECG) and echocardiography (ECHO). Results: Liraglutide treatment significantly attenuated oxidative stress, repaired ECG and ECHO parameters, and mitigated myocardial morphological changes induced by isoprenaline. Conclusions: Liraglutide restores cardiac function in isoprenaline-induced HF.

## 1. Introduction

Myocardial injury (MI) is characterized by at least one of the cardiac troponin levels exceeding the 99th percentile of the upper reference limit [1]. Generally, a relative increase of 20% in troponin concentration is considered to be significant enough for diagnosing MI [2,3]. The term myocardial injury encompasses a wide range of conditions. The key difference between myocardial injury and myocardial infarction is determined by the source of elevated troponin. MI refers to the death of myocardial cells from nonischemic causes, whereas myocardial infarction is linked to ischemic factors, including plaque disruption and compromised oxygen supply to the myocardium [4]. The release of cardiac troponin into the bloodstream is associated with cardiomyocyte necrosis and apoptosis, an increase in membrane permeability, and mechanical stretching from pressure or volume overload without myocardial ischemia [5]. It is acknowledged that numerous conditions can lead to MI, such as tachyarrhythmia, myocarditis, Takotsubo syndrome, and valvular heart disease [2].

MI is marked by oxidative stress and the activation of the inflammatory cytokine response [6]. Accumulation of reactive oxygen species (ROS) in the mitochondria can result in a harmful cycle that causes damage to mitochondrial DNA and a reduction in mitochondrial functionality, subsequently leading to the production of more ROS. These ROS can promote cardiomyocyte hypertrophy, cardiomyocyte apoptosis, and interstitial fibrosis through the activation of matrix metalloproteinases (MMPs), thus perpetuating a detrimental cycle of adverse ventricular remodeling and the progression to heart failure (HF) [7,8]. MI initiates an inflammatory response designed to eliminate necrotic cellular debris and commence the process of anti-inflammatory repair. The occurrence of cellular necrosis and apoptosis is crucial for the activation of the inflammatory immune system. This activation has a dual purpose: while the chronic inflammatory state can lead to maladaptive remodeling, the infiltration of inflammatory cells has the capacity to recruit leukocytes, clear tissue debris, and initiate the healing response, ultimately facilitating the formation of scar tissue [8]. There is a reciprocal relationship between oxidative stress and inflammatory cytokines: oxidative stress promotes inflammation, which in turn increases oxidative stress. This interplay exacerbates changes in the heart after an MI, affecting cardiac function and prognosis [6]. All of these processes can lead to HF [9]. Notably, this progression to HF can occur even while the patient is still receiving treatment and care in a hospital setting. It is noteworthy that 75% of individuals affected by MI die within five years following the event [5]. This highlights the urgent need for effective prevention and early intervention for individuals at risk of or recovering from MI.

In the context of translational medicine, there is an ongoing pursuit of suitable experimental models that accurately replicate human diseases. As a result, isoprenaline is commonly used in studies of MI [10]. The myocardial injury caused by isoprenaline is distinguished by the presence of oxidative stress and an inflammatory response [11,12].

Many drugs and procedures have been used in order to prevent the development of HF following MI. Liraglutide is an established GLP-1 receptor agonist (GLP-1 RA) that is commonly used in the treatment of diabetes. The use of GLP-1 RAs is supported by their beneficial effects on glucose metabolism and the functioning of the pancreas. These agents not only enhance the survival of pancreatic cells but also decrease glycemic levels. Given the serious complications that can arise from diabetes and its various comorbidities, the therapeutic effects of GLP-1 RAs are being studied in a wide range of diseases and conditions associated with diabetes. There is evidence that GLP-1 RAs can also improve neurological impairments, including cognitive decline and Alzheimer’s disease [13], dermatological conditions such as psoriasis [14], and renal diseases [15]. Recent studies have revealed that liraglutide exhibits beneficial effects, including its role in reducing inflammation [16] and oxidative stress [17].

We have recently established that pretreatment with liraglutide has significant cardioprotective effects in acute MI [18]. Based on the previous study, we aimed to evaluate the potential role of liraglutide in the treatment of MI and in the prevention of HF. For this purpose, the rats were treated with two consecutive doses of isoprenaline followed by liraglutide treatment for the next seven days.

## 2. Materials and Methods

### 2.1. Experimental Animals

In this study, male Wistar albino rats were used. The animals were kept under specific laboratory conditions, with a room temperature of 21 ± 2 °C, humidity level of 55 ± 5%, and a light and dark cycle of 12 h each, commencing at 08:00 a.m. This research, along with all procedures, protocols, and experimental animals, was approved by the Ethics Committee for the Protection of Welfare of Experimental Animals at the Faculty of Medicine, University of Banja Luka (approval number 18/1.190-13/22, dated 1 June 2022). The housing of the animals adhered to the guidelines established by the National Institute of Health (NIH) for the care and use of laboratory animals.

### 2.2. Experimental Grouping

The animals were categorized into four distinct groups. Over a period of 7 days, they received either 0.9% of NaCl (saline) or liraglutide treatment after isoprenaline-induced MI. The control group (Con, n = 6) was administered 1 mL/kg of 0.9% NaCl subcutaneously (s.c.) on the first 2 days, followed by 1 mL/kg of 0.9% NaCl s.c. for the subsequent 7 days. The isoprenaline group (Iso, n = 8) was given isoprenaline at a dosage of 85 mg/kg dissolved in 1 mL/kg of saline, administered s.c. on the first 2 days, and then received 1 mL/kg of 0.9% NaCl s.c. for the next 7 days. The liraglutide group (Lir, n = 6) was treated with 0.9% NaCl s.c. on the first 2 days, followed by liraglutide at a dosage of 1.8 mg/kg s.c. for 7 days. Lastly, the liraglutide + isoprenaline group (Iso + Lir, n = 8) received isoprenaline at 85 mg/kg s.c. on the first 2 days, followed by liraglutide at 1.8 mg/kg subcutaneously for the remaining 7 days (Figure 1).

### 2.3. Obtaining Blood and Tissue Samples

At the end of the experiment, all rats were anesthetized, and blood samples were drawn from the aorta into Vacutainer tubes intended for serum and plasma citrate. To obtain serum, the blood samples were left at room temperature for 20 min to allow clotting, followed by centrifugation at 3000 rpm for 5 min. Plasma samples underwent centrifugation at 3000 rpm for 10 min. After separating the plasma, red blood cells were washed three times with three volumes of cold 0.9% NaCl. All samples were preserved at −80 °C until further analysis. The hearts were removed and placed in small plastic containers filled with 10% formalin for histological examination.

### 2.4. Biomarkers of Myocardial Injury and Other Biochemical Parameters

The concentration of serum high sensitive troponin I (hs TnI) was assessed on the Abbot Alinity ci-series platform utilizing chemiluminescent microparticle immunoassay (CMIA). Measurements of serum glucose, level of lipids (total cholesterol—TC, high-density lipoprotein—HDL, low-density lipoprotein—LDL, triglycerides—TG), and the activities of aspartate aminotransferase (AST), alanine aminotransferase (ALT), and lactate dehydrogenase (LDH) were conducted using the same Abbot Alinity ci-series through chemiluminescence immunoassay (CLIA). N-terminal pro-brain natriuretic peptide (NT-proBNP) was measured by enzyme-linked immunosorbent assay (ELISA) with the FineTest Rat NT-proBNP (N-Terminal Pro-Brain Natriuretic Peptide) ELISA Kit, Wuhan Fine Biotech Co., Ltd. Wuhan, China.

### 2.5. Prooxidative and Antioxidative Markers

Prooxidative markers, including thiobarbituric acid reactive substances (TBARS), superoxide anion radical (O_2_^−^), and hydrogen peroxide (H_2_O_2_), and nitrite (NO_2_^−^), were assessed in plasma. The TBARS level, indicative of lipid peroxidation, was quantified using 1% TBA and 0.05 M sodium hydroxide (NaOH), measured at wavelength of 530 nm. The plasma concentration of O_2_^−^ was evaluated by using nitro blue tetrazolium (NTB) in conjunction with the TRIS buffer, with a measurement also of 530 nm. The concentration of plasma H_2_O_2_ was determined using the Pick and Keisari method, which relies on the oxidation of phenol red by H_2_O_2_, with measurements taken at a wavelength of 610 nm. For the determination of nitrite levels, the Green method was employed, utilizing 30% sulfosalicylic acid and Griess reagent [19]. The antioxidative markers catalase (CAT) and superoxide dismutase (SOD), along with reduced glutathione (GSH) levels, were analyzed in red blood cell lysates according to Beutler methods, with results obtained through spectrophotometric measurement [19].

### 2.6. ECG Recording

ECG recordings were conducted on three separate occasions. The initial two recordings took place on the first day of the experiment, specifically before and 10 min following the administration of the first dose of isoprenaline. The third recording was executed at the end of the experiment, before the sacrifice of the animals, on day 10. Prior to the recordings, the rats were anesthetized using a combination of ketamine (30 mg/kg) and xylazine (5 mg/kg) [20]. The ECGs were captured with a sensitivity setting of 2 cm per 1 mV and a speed of paper of 25 mm/s. Lead II was used to analyze the heart rate (beats per minute, bpm), QT interval (seconds, s), and QRS peak-to-peak voltage amplitude (millivolts, mV) [21] in each tracing.

### 2.7. ECHO

On the tenth day of the experiment, all rats were subjected to anesthesia via intraperitoneal administration of 30 mg/kg ketamine and 5 mg/kg xylazine An ECHO was conducted following the ECG. Transthoracic two-dimensional (2D) echocardiography was performed using a Logio 400 CL ultrasound device with an 11 MHz phased array transducer to assess cardiac structure and function. M-mode echocardiographic images were captured in parasternal long-axis and short-axis views at the papillary muscle tips. Measurements included systolic and diastolic septal (IVSs and IVSd) and posterior wall thickness (PDWs, PDWd) and left ventricular internal diameters (LVIDs and LVIDd). Left ventricular ejection fraction (EF), fractional shortening (FS), and end-systolic (ESV) and end-diastolic volume (EDV) were calculated [22]. All measurements were conducted by the same observer following the American Society of Echocardiography guidelines [23,24].

### 2.8. Pathohistological Findings

Isolated hearts were initially preserved in 10% formalin, followed by the formation of tissue blocks using paraffin wax. Each block was subsequently sectioned into 4 µm slices using a microtome and stained with hematoxylin and eosin (H&E). Myocardial injuries were assessed and assigned a score ranging from 1 to 4. A score of 1 indicates no pathological changes in the myocardium; a score of 2 reflects mild damage characterized by multifocal degeneration and slight inflammatory infiltration or localized damage of cardiomyocytes; a score of 3 denotes moderate damage, featuring significant degeneration of cardiomyocytes and/or widespread inflammation; and a score of 4 signifies severe damage, including necrosis accompanied by diffuse inflammation [18,25]. The heart slices were evaluated, and the average score for each group was computed. The assessment of myocardial fibrosis is done by Masson’s trichrome stain, which distinctly marks the deposition of collagen within the myocardial tissue [26].

### 2.9. Statistical Analysis

Statistical analysis was conducted utilizing IBM-SPSS Statistics version 29.0 software (SPSS, Inc., Chicago, IL, USA). The ANOVA test was employed to assess the means of parametric variables. For nonparametric variables, the Kruskal–Wallis and Mann–Whitney U tests were used to compare differences among groups. Post hoc analysis was carried out using Tukey, Bonferroni, and LSD tests. The results are expressed as mean ± standard error, with a *p*-value of less than 0.05 deemed statistically significant.

## 3. Results

### 3.1. Effects of Liraglutide Treatment on Body Weight (BW), Heart Weight (HW), and Heart-to-Body Weight (H/BW) Ratio in Rats with Isoprenaline-Induced HF

Liraglutide significantly reduced body weight (*p* < 0.05 versus Con), and the combination of isoprenaline and liraglutide further enhanced this body weight-loss effect. Ten days after the first dose of isoprenaline, heart weight and size increased (Table 1, Figure 2). The H/BW ratio is often used as a more accurate measure of heart size and of heart hypertrophy, and it was increased in isoprenaline-treated rats. However, liraglutide was able to alleviate this alteration.

### 3.2. Effects of Liraglutide Treatment on Biomarkers of MI and Other Biochemical Parameters in Rats with Isoprenaline-Induced HF

An elevation in cardiac troponin levels is indicative of MI, and elevation in NT-proBNP of HF. A notable rise in the levels of these markers was observed ten days after the initial dose of isoprenaline, and the administration of liraglutide effectively mitigated that increase (Figure 3a,b).

Enzymes related to MI, including AST and LDH, were also assessed. Ten days after the first dose of isoprenaline, activities of these enzymes were elevated. Furthermore, ALT activity was measured and revealed a significant rise (*p* < 0.05 Iso versus Con). Moreover, treatment with liraglutide effectively reduced the activities of these enzymes (Table 2).

The administration of isoprenaline, liraglutide, and their combination decreased glucose levels.

The administration of isoprenaline induced a significant change in lipid profile; it increased serum TC, HDL, LDL, and TG (Table 3). The administration of liraglutide significantly restored the levels of TC and HDL (*p* < 0.05 Iso + Lir versus Iso).

### 3.3. Effects of Liraglutide Treatment on Prooxidative and Antioxidative Markers in Rats with Isoprenaline-Induced HF

A significant rise in TBARS, O_2_^−^, and H_2_O_2_ (*p* < 0.05 Con versus Iso) in rats with isoprenaline-induced HF indicates a substantial level of oxidative stress (Figure 4a–d). However, these prooxidative markers were reduced in rats treated with liraglutide. Furthermore, there was a decrease in NO_2_^−^ level, which was alleviated by liraglutide treatment.

Following the first dose of isoprenaline, decreased activities of CAT, SOD, and the level of GSH were recorded ten days later. However, liraglutide managed to restore the reduced levels (*p* < 0.05 Iso + Lir versus Iso) of CAT, SOD, and GSH (Figure 4e–g).

### 3.4. Effects of Liraglutide Treatment on Electrocardiogram (ECG) in Rats with Isoprenaline-Induced HF

Isoprenaline administration resulted in a significant increase in heart rate—HR (*p* < 0.001 versus Con)—following the initial injection in both the Iso and Iso + Lir groups, as these groups were identical at that stage (Table 4). Additionally, isoprenaline caused an increase in QRS amplitude following the first dose (*p* < 0.05 Iso versus Con, *p* < 0.05 Iso + Lir versus Con) and a decrease by the end of the experiment (*p* < 0.05 versus Con). After the first dose of isoprenaline, all rats exhibited negative T waves, but ten days later, on the day of sacrifice, the T waves in all animals were positive.

### 3.5. Effects of Liraglutide Treatment on Echocardiogram (ECHO) in Rats with Isoprenaline-Induced HF

Ten days after the first dose of isoprenaline, the left ventricle internal diameter in systole (LVIDs) was increased (Figure 5). The treatment with liraglutide effectively inhibited the rise of LVIDs (*p* < 0.05 Iso + Lir versus Iso).

The thickness of the interventricular septum showed a reduction ten days after the administration of the first dose of isoprenaline, with a more pronounced effect observed during systole compared to diastole (Figure 6a,b); however, this change did not achieve statistical significance. Treatment with liraglutide appeared to mitigate this change.

The administration of isoprenaline also influenced various echocardiographic parameters related to systolic function, including an increase in ESV)and a decrease in PDWs, EF, and FS (*p* < 0.05 versus Con, Figure 6c,e,g,h). Treatment with liraglutide alleviated these modifications (*p* < 0.05 versus Iso + Lir versus Iso).

The relationship of NT-proBNP with ECHO parameters is shown in Figure 7. A significant negative correlation was found between NT-proBNP and PWDd, as well as between NT-proBNP and PWDs. Furthermore, NT-proBNP exhibited a positive correlation with LVIDs and ESV, while it showed negative correlations with both EF and FS.

### 3.6. Effects of Liraglutide Treatment on Myocardium Morphology in Rats with Isoprenaline-Induced HF

Ten days after the administration of an initial dose of isoprenaline, considerable damage to the myocardium was observed. It was characterized by prominent fragmentation of cardiomyocytes, intercellular edema, infiltration of leukocytes, and hemorrhaging (Figure 8b). A seven-day liraglutide treatment markedly reduced these morphological alterations (Figure 8d,e).

Figure 9 shows the extent of collagen deposition in myocardium ten days after the first dose of isoprenaline.

## 4. Discussion

Isoprenaline is recognized as a model for MI because it provides adequate mimicry and maintains a low mortality rate, reflecting its considerable reliability in translational medicine [10,27]. The administration of isoprenaline has been associated with heart enlargement [28], represented by the rise in H/BW in this study. Similar observations were made in other studies [29,30].

The findings from this study indicated a reduction in body weight among rats treated with isoprenaline and liraglutide. The observed weight loss in the isoprenaline-treated group may be attributed to the activation of β-adrenoceptors, leading to a decrease in plasma leptin levels, potentially mediated by an increase in cyclic AMP (cAMP) within the adipose tissue. Catecholamines are significant lipolytic hormones, primarily acting through β-adrenoceptors [31]. Liraglutide, a GLP-1 receptor agonist, is used not only for the management of diabetes but also for the treatment of obesity. It demonstrates a considerable impact on body weight reduction [32]. This study noted similar effects of liraglutide on weight loss, and it was particularly pronounced in rats treated with both liraglutide and isoprenaline.

The diagnosis of MI was substantiated by an elevated level of hs TnI. The results of other studies involving the isoprenaline model of MI demonstrated an increase in cardiac troponins [10,33,34,35]. Natriuretic peptides are recognized as significant biomarkers for diagnosing HF, assessing its severity, and predicting outcomes, as well as potentially aiding in its management. B-type natriuretic peptide (BNP) and NT-proBNP are considered indicators of ventricular stretch, produced in response to the wall stress. As a result, these peptides have become essential components in the evaluation of HF, with their concentration generally increasing in correlation with the deterioration of the condition [36]. An increase in NT-proBNP levels suggests the development of HF. Liraglutide treatment was associated with a significant reduction in hs TnI and NT-proBNP levels, emphasizing the important function of liraglutide in the prevention of HF.

Liraglutide reduces blood glucose by stimulating glucose-dependent insulin secretion and inhibiting glucagon. It effectively lowers glycated hemoglobin and glycemia in adults and may improve pancreatic beta cell function [37]. This study’s results demonstrated a lower glucose concentration in animals receiving liraglutide treatment. Additionally, rats treated with isoprenaline also exhibited a decrease in glucose levels, consistent with the observations made by Sobot et al. [10]. Results from the in vitro study of Fiserova et al. [38] demonstrated that acute exposure of cardiomyocytes to isoprenaline led to an elevation in glucose flux at the high dosage of isoprenaline. However, an extended exposure to isoprenaline did not produce a significant impact on glucose flux. The research conducted by Chang et al. [39] indicated that the administration of isoprenaline in obese rats led to an exacerbation of hyperglycemia. It is important to observe that the impact of isoprenaline on glucose metabolism is influenced by both the dosage and the duration of exposure.

The study also revealed an increase in AST, LDH, and ALT enzymes in rats treated with isoprenaline, but significance was noted only for ALT. The findings from multiple studies suggest that the rise in these enzymes is characteristic of the isoprenaline model of MI [11,34,35,40,41,42,43,44]. The activities of LDH, AST, and ALT serve as indicators of changes in the integrity and permeability of cardiomyocyte membranes. An elevation in these cytosolic enzymes in the plasma points to a leakage into the bloodstream, likely resulting from an increased permeability or damaged membranes [35]. Seven-day liraglutide treatment corrected the alterations observed in the activity of these enzymes, but not significantly. The meta-analysis conducted by Malik et al. [45] did not show a statistically significant effect of liraglutide on the reduction of ALT and AST.

This study indicated a rise in TC, LDL, HDL, and TG in rats treated with isoprenaline; however, the rise was significant only in TC and HDL levels. The observed increases in TC, LDL, and TG align with findings from other studies examining isoprenaline-induced MI [46,47,48,49]. While Sudha et al. [46] reported an increase in HDL levels, Galal et al. [47] documented a decrease in HDL. Treatment with liraglutide improved the lipid profile in isoprenaline-treated rats, significantly in TC and HDL levels. Similar results were noted in clinical studies [45,50,51].

Increased levels of TBARS, H_2_O_2_, and O_2_^−^ suggest that isoprenaline induced significant oxidative stress. Similar results have been found by other researchers [11,43,49]. In contrast, the level of NO_2_^−^ decreased, which is supported by the study of Rankovic et al. [52], but in a different model of MI, a doxorubicin model. Oxidative stress, along with the generation of oxygen-derived free radicals, is considered a fundamental cause of the various processes that lead to myocardial damage induced by isoprenaline [43]. The presence of highly reactive radicals can lead to lipid peroxidation and initiate cell death through various mechanisms, including apoptosis and autophagy [43,53]. The findings of this study indicated a decline in the activities of SOD and CAT, as well as a reduction in GSH levels, in rats treated with isoprenaline. These findings align with the results of other studies [11,12,35,40,43,49,54]. However, liraglutide demonstrated significant antioxidative properties in both healthy rats and those with isoprenaline-induced HF. The administration of liraglutide resulted in elevated SOD and CAT activities as well as GSH level. Results of the experimental studies showed antioxidative effects [55,56] of the liraglutide treatment similar to ours.

In the current study, an ECG was conducted to assess cardiac function, and to, alongside cardiac troponin levels, confirm the occurrence of MI. Following the administration of isoprenaline, there was a notable increase in heart rate and consequently a decrease in QT interval, which aligns with expectations of isoprenaline as a β-adrenergic agonist. This observation is consistent with findings from other researchers [40]. In addition to the elevated heart rate and the presence of negative T waves, which support the diagnosis of MI [57], alterations in the peak-to-peak amplitude of the QRS complex were observed. The QRS amplitude was elevated in rats treated with isoprenaline, indicating modifications in ventricular depolarization and suggesting ventricular dysfunction. The administration of liraglutide resulted in improvements in electrocardiographic changes.

The ECHO analysis demonstrated that isoprenaline significantly affected cardiac function, with a predominant impact on systolic activity. This influence is characterized by an increase in LVIDs, ESV, and EDV, and a reduction in IVSs, IVSd, PWDs, PWDd, EF, and FS. The alterations in LVIDs, IVs, PDWs, ESV, EF, and FS were significant, suggesting that isoprenaline has a more pronounced effect on systolic function. Similar conclusions were drawn by Li et al. [34]. Decreased EF and FS highlight the substantial impact of isoprenaline on cardiac performance [26,36,58]. Some studies have reported increases in LVIDd, PWDs, and PWDd [36,48]. These differences might result from the distinct experimental protocols and the varying doses of isoprenaline, or the extent of remodeling of the left ventricle. The geometric changes occurring in the ventricle, such as those of shape and size, are the primary driving forces behind the remodeling of the left ventricle. In the early stages of MI, there is an increase in cardiac load as a compensatory response to mechanical and physiological stress. This leads to cardiomyocyte hypertrophy, resulting in myocardial hypertrophy. The decline in cardiac systolic function, along with a subsequent increase in LV volume, can elevate wall stress and oxygen requirements, ultimately contributing to maladaptive tissue remodeling and the onset of HF [8]. The findings of this study indicate that liraglutide has the potential to restore compromised cardiac function and prevent the onset of HF.

Pathohistological assessment showed that administration of isoprenaline resulted in considerable myocardial damage, reflected by a heightened myocardial damage score. The presence of granulocyte infiltration signifies the initiation of myocardial repair after isoprenaline-induced MI. Collagen accumulation indicates the beginning of the heart’s remodeling phase [8]. Similar results were presented in other studies [26,27,44,49]. Liraglutide reduced changes in the myocardium induced by isoprenaline, such as inflammation, interstitial edema, myofibrillar degeneration, and fibrosis.

## 5. Conclusions

Liraglutide administration diminishes markers associated with MI and HF, such as hs TnI and NT-proBNP, and facilitates the repair of MI induced by isoprenaline. Furthermore, liraglutide therapy mitigates oxidative stress by lowering levels of TBARS, H_2_O_2_, and O_2_^−^, while simultaneously increasing the activity of CAT and SOD as well as GSH levels. It also restores impaired systolic cardiac function by reducing LVIDs and enhancing EF and FS, effectively preventing HF. Pathohistological examinations reveal that liraglutide effectively reduces inflammation and fibrosis, thereby inhibiting heart remodeling and HF in isoprenaline-treated rats.

## Figures and Tables

**Figure 1 life-15-00443-f001:**
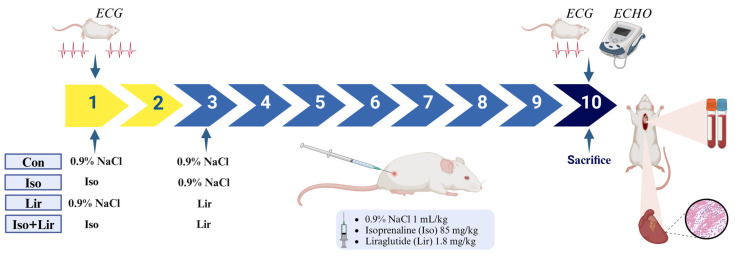
Study design: animal grouping and study protocol. Con—0.9% NaCl s.c. on the first 2 days + 0.9% NaCl s.c. for the next 7 days; Iso—isoprenaline 85 mg/kg s.c. on the first 2 days + 0.9% NaCl s.c. for the next 7 days; Lir—0.9% NaCl s.c. on the first 2 days + liraglutide 1.8 g/kg for the next 7 days; Iso + Lir—isoprenaline 85 mg/kg s.c. on the first 2 days + liraglutide 1.8 g/kg for the next 7 days.

**Figure 2 life-15-00443-f002:**
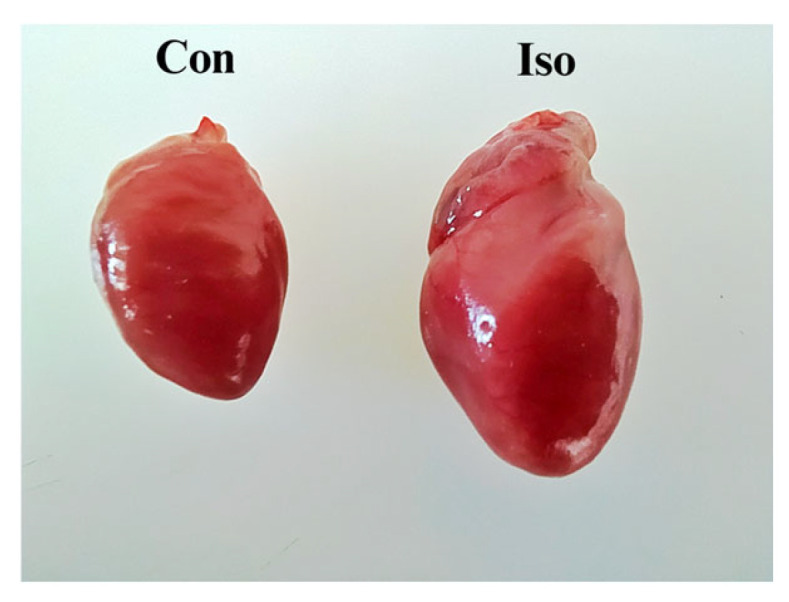
Isoprenaline-induced cardiac hypertrophy: comparison of normal (Con group) and isoprenaline-treated heart (Iso group).

**Figure 3 life-15-00443-f003:**
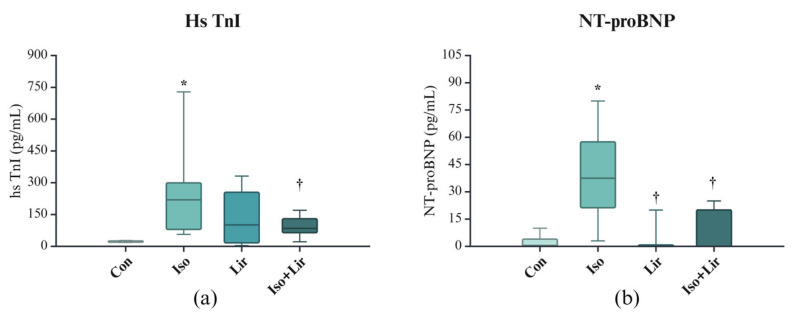
Markers of myocardial damage: (**a**) high sensitive troponin I (hs TnI) and (**b**) N-terminal pro-brain natriuretic peptide (NT-proBNP). Con—0.9% NaCl s.c. on the first 2 days + 0.9% NaCl s.c. for the next 7 days; Iso—isoprenaline 85 mg/kg s.c. on the first 2 days + 0.9% NaCl s.c. for the next 7 days; Lir—0.9% NaCl s.c. on the first 2 days + liraglutide 1.8 g/kg for the next 7 days; Iso + Lir—isoprenaline 85 mg/kg s.c. on the first 2 days + liraglutide 1.8 g/kg for the next 7 days; * *p* < 0.05 versus Con, † *p* < 0.05 versus Iso.

**Figure 4 life-15-00443-f004:**
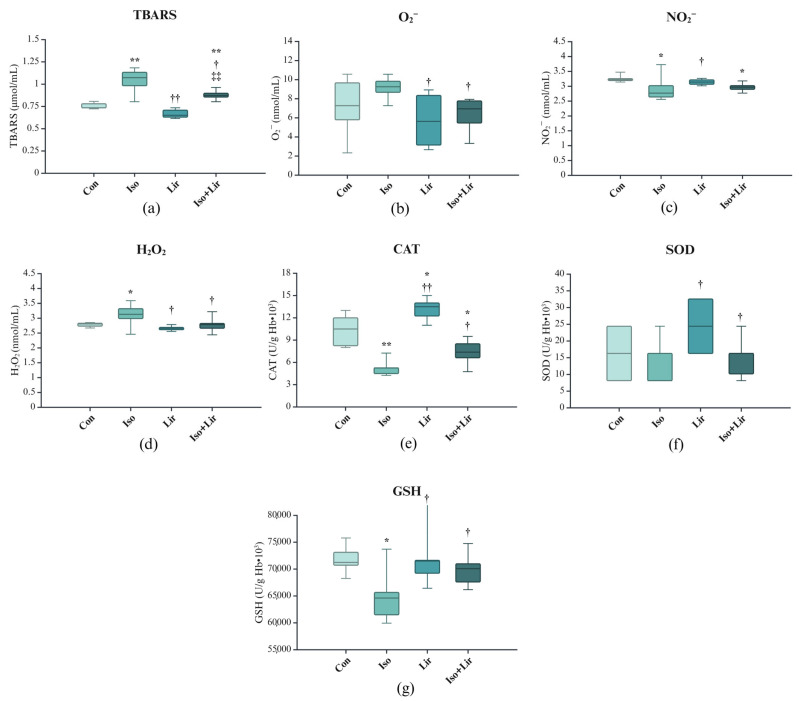
Effect of liraglutide treatment on oxidative stress markers in rats with isoprenaline-induced HF: (**a**) thiobarbituric acid reactive substances (TBARS); (**b**) superoxide anion radical (O_2_^−^); (**c**) nitrite (NO_2_^−^); (**d**) hydrogen peroxide (H_2_O_2_); (**e**) catalase (CAT); (**f**) superoxide dismutase (SOD); (**g**) reduced glutathione (GSH). Con—0.9% NaCl s.c. on the first 2 days + 0.9% NaCl s.c. for the next 7 days; Iso—isoprenaline 85 mg/kg s.c. on the first 2 days + 0.9% NaCl s.c. for the next 7 days; Lir—0.9% NaCl s.c. on the first 2 days + liraglutide 1.8 g/kg for the next 7 days; Iso + Lir—isoprenaline 85 mg/kg s.c. on the first 2 days + liraglutide 1.8 g/kg for the next 7 days; * *p* < 0.05 versus Con, ** *p* < 0.001 versus Con, † *p* < 0.05 versus Iso, †† *p* < 0.001 versus Iso, ‡‡ *p* < 0.001 versus Lir.

**Figure 5 life-15-00443-f005:**
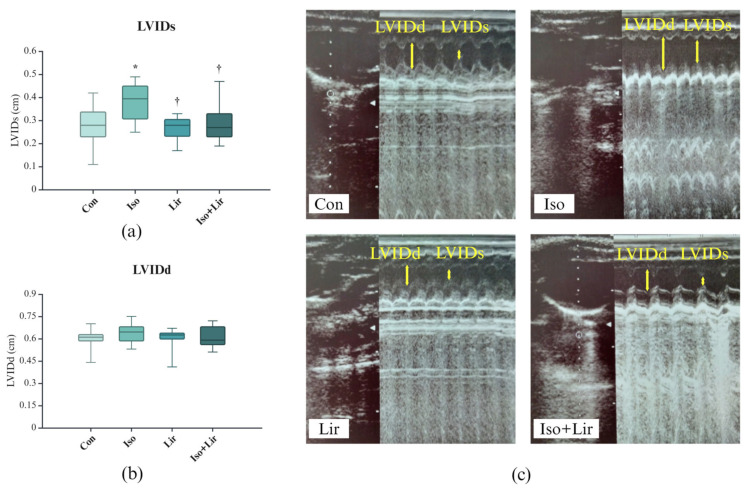
Effect of liraglutide treatment on left ventricle internal diameter (LVID) in rats with isoprenaline-induced HF: (**a**) in systole (LVIDs); (**b**) in diastole (LVIDd); (**c**) ECHO image of LVIDs and LVIDd (marked with yellow arrows) in all groups. Con—0.9% NaCl s.c. on the first 2 days + 0.9% NaCl s.c. for the next 7 days; Iso—isoprenaline 85 mg/kg s.c. on the first 2 days + 0.9% NaCl s.c. for the next 7 days; Lir—0.9% NaCl s.c. on the first 2 days + liraglutide 1.8 g/kg for the next 7 days; Iso + Lir—isoprenaline 85 mg/kg s.c. on the first 2 days + liraglutide 1.8 g/kg for the next 7 days; * *p* < 0.05 versus Con, † *p* < 0.05 versus Iso.

**Figure 6 life-15-00443-f006:**
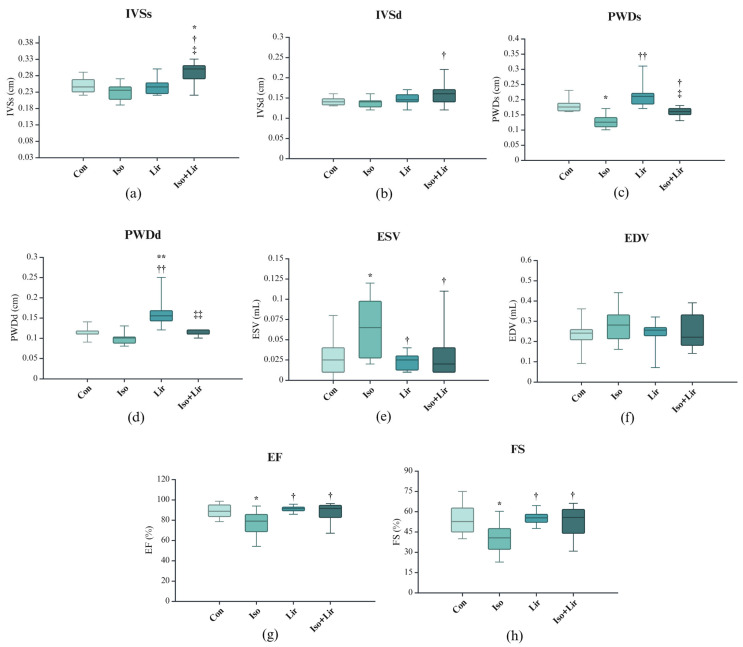
Effect of liraglutide treatment on ECHO parameters in rats with isoprenaline-induced HF: (**a**) interventricular septum thickness in systole (IVSs); (**b**) interventricular septum thickness in diastole (IVSd); (**c**) posterior wall diameter in systole (PWDs); (**d**) posterior wall diameter in diastole (PWDd); (**e**) end-systolic volume; (**f**) end-diastolic volume (EDV); (**g**) ejection fraction (EF); (**h**) fractional shortening (FS). Con—0.9% NaCl s.c. on the first 2 days + 0.9% NaCl s.c. for the next 7 days; Iso—isoprenaline 85 mg/kg s.c. on the first 2 days + 0.9% NaCl s.c. for the next 7 days; Lir—0.9% NaCl s.c. on the first 2 days + liraglutide 1.8 g/kg for the next 7 days; Iso + Lir—isoprenaline 85 mg/kg s.c. on the first 2 days + liraglutide 1.8 g/kg for the next 7 days; * *p* < 0.05 versus Con, ** *p* < 0.001 versus Con, † *p* < 0.05 versus Iso, †† *p* < 0.001 versus Iso, ‡ *p* < 0.05 versus Lir, ‡‡ *p* < 0.001 versus Lir.

**Figure 7 life-15-00443-f007:**
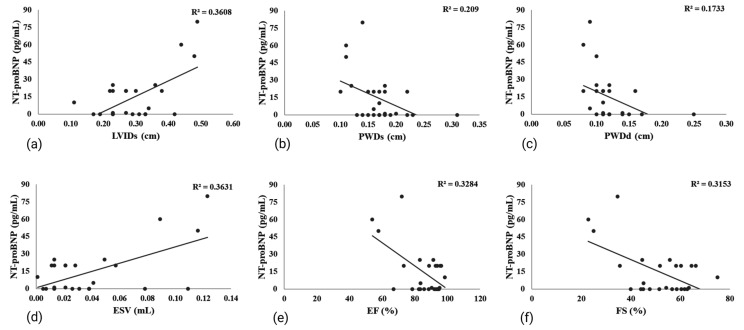
The relationship of NT-proBNP with ECHO parameters: (**a**) LVIDs (Pearson correlation R = 0.601, *p* < 0.05); (**b**) PWDs (Pearson correlation R = −0.416, *p* < 0.05); (**c**) PWDs (Pearson correlation R = −0.457, *p* < 0.05) (**d**) ESV (Pearson correlation R = 0.603, *p* < 0.05); (**e**) EF (Pearson correlation R = −0.573, *p* < 0.05); (**f**) FS (Pearson correlation R = −0.562, *p* < 0.05).

**Figure 8 life-15-00443-f008:**
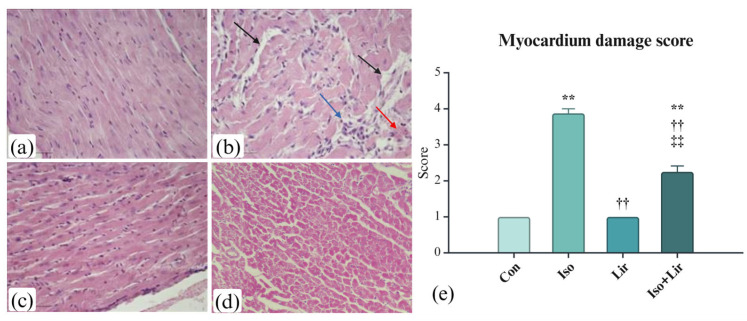
Histological characteristics of myocardium after the liraglutide treatment in rats with isoprenaline-induced HF (H&E, magnification ×20): (**a**) Con group—normal histological morphology; (**b**) Iso group—interstitial edema (black arrow), bleeding (red arrow), inflammation (blue arrow); (**c**) Lir group—normal histological morphology; (**d**) Iso + Lir group—reduction of morphological changes caused by isoprenaline; (**e**) myocardium damage score. Con—0.9% NaCl s.c. on the first 2 days + 0.9% NaCl s.c. for the next 7 days; Iso—isoprenaline 85 mg/kg s.c. on the first 2 days + 0.9% NaCl s.c. for the next 7 days; Lir—0.9% NaCl s.c. on the first 2 days + liraglutide 1.8 g/kg for the next 7 days; Iso + Lir—isoprenaline 85 mg/kg s.c. on the first 2 days + liraglutide 1.8 g/kg for the next 7 days; ** *p* < 0.001 versus Con, †† *p* < 0.001 versus Iso, ‡‡ *p* < 0.001 versus Lir.

**Figure 9 life-15-00443-f009:**
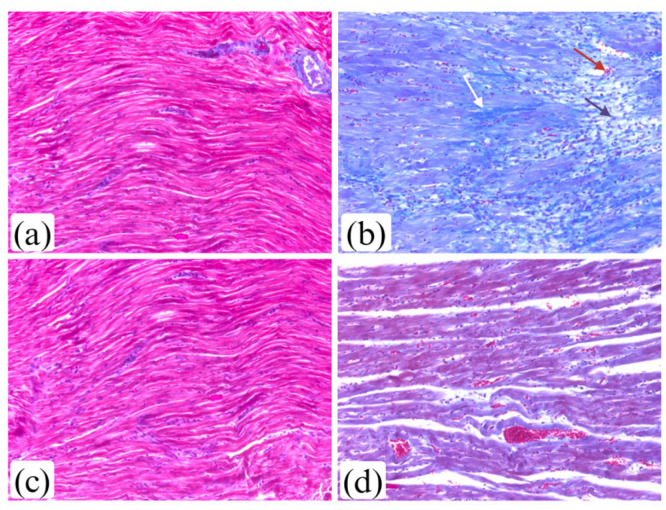
The extent of cardiac fibrosis after the liraglutide treatment in rats with isoprenaline-induced HF (Masson trichrome stain, magnification ×20): (**a**) Con group—normal morphology of myocardium; (**b**) Iso group—fibrosis (blue color) with collagen deposits (white arrow), inflammation (purple arrow), bleeding (red arrow), myofibrillar degeneration; (**c**) Lir group—normal morphology of myocardium; (**d**) Iso + Lir group—mild myofibrillar degeneration, dilated blood vessels with extravasation of red blood cells, without inflammation and fibrosis.

**Table 1 life-15-00443-t001:** Effects of liraglutide treatment on BW, HW, and H/BW in rats with isoprenaline-induced HF.

Groups (Mean ± SE)				
Parameters	Con	Iso	Lir	Iso + Lir
BW (g)	261.33 ± 10.78	228.33 ± 6.31	216.33 ± 8.16 *	199.56 ± 8.86 ** †
HW (g)	0.80 ± 0.02	0.82 ± 0.02	0.71 ± 0.03 †	0.68 ± 0.03 * †
H/BW ratio	3.05 ± 0.07	3.62 ± 0.09 *	3.22 ± 0.06 †	3.40 ± 0.09 *

BW—body weight; HW—heart weight; H/BW—heart-to-body weight ratio; Con—0.9% NaCl s.c. on the first 2 days + 0.9% NaCl s.c. for the next 7 days; Iso—isoprenaline 85 mg/kg s.c. on the first 2 days + 0.9% NaCl s.c. for the next 7 days; Lir—0.9% NaCl s.c. on the first 2 days + liraglutide 1.8 g/kg for the next 7 days; Iso + Lir—isoprenaline 85 mg/kg s.c. on the first 2 days + liraglutide 1.8 g/kg for the next 7 days; * *p* < 0.05 versus Con, ** *p* < 0.001 versus Con, † *p* < 0.05 versus Iso.

**Table 2 life-15-00443-t002:** Effect of liraglutide treatment on serum biochemical parameters in rats with isoprenaline-induced HF.

Groups (Mean ± SE)				
Parameters	Con	Iso	Lir	Iso + Lir
AST (U/L)	250.58 ± 39.95	363.78 ± 41.80	284.50 ± 36.84	350.67 ± 49.68
ALT (U/L)	108.67 ± 32.53	130.11 ± 7.94 *	80.17 ± 12.46 †	123.83 ± 18.64
LDH (U/L)	1202.67 ± 355.89	1968.67 ± 299.43	1491.00 ± 264.07	1916.67 ± 399.24
Glucose (mmol/L)	24.58 ± 3.02	17.07 ± 2.47 *	22.38 ± 1.62	21.30 ± 1.01

AST—aspartate aminotransferase; ALT—alanine aminotransferase; LDH—lactate dehydrogenase; Con—0.9% NaCl s.c. on the first 2 days + 0.9% NaCl s.c. for the next 7 days; Iso—isoprenaline 85 mg/kg s.c. on the first 2 days + 0.9% NaCl s.c. for the next 7 days; Lir—0.9% NaCl s.c. on the first 2 days + liraglutide 1.8 g/kg for the next 7 days; Iso + Lir—isoprenaline 85 mg/kg s.c. on the first 2 days + liraglutide 1.8 g/kg for the next 7 days; * *p* < 0.05 versus Con, † *p* < 0.05 versus Iso.

**Table 3 life-15-00443-t003:** Effect of liraglutide treatment on serum lipid profile in rats with isoprenaline-induced HF.

Groups (Mean ± SE)				
Parameters	Con	Iso	Lir	Iso + Lir
TC (mmol/L)	1.00 ± 0.08	1.80 ± 0.23 *	0.75 ± 0.02 * ††	1.13 ± 0.17 † ‡
HDL (mmol/L)	0.37 ± 0.03	0.69 ± 0.07 *	0.28 ± 0.02 ††	0.46 ± 0.07 †
LDL (mmol/L)	0.16 ± 0.02	0.19 ± 0.03	0.10 ± 0.00 †	0.14 ± 0.02
TG (mmol/L)	0.75 ± 0.15	1.03 ± 0.45	0.66 ± 0.41	0.88 ± 0.16

TC—total cholesterol; HDL—high-density lipoprotein; LDL—low-density lipoprotein; TG—triglycerides; Con—0.9% NaCl s.c. on the first 2 days + 0.9% NaCl s.c. for the next 7 days; Iso—isoprenaline 85 mg/kg s.c. on the first 2 days + 0.9% NaCl s.c. for the next 7 days; Lir—0.9% NaCl s.c. on the first 2 days + liraglutide 1.8 g/kg for the next 7 days; Iso + Lir—isoprenaline 85 mg/kg s.c. on the first 2 days + liraglutide 1.8 g/kg for the next 7 days; * *p* < 0.05 versus Con, † *p* < 0.05 versus Iso, †† *p* < 0.001 versus Iso, ‡ *p* < 0.05 versus Lir.

**Table 4 life-15-00443-t004:** Effects of liraglutide treatment on ECG parameters in rats with isoprenaline-induced HF.

Groups (Mean ± SE)				
Parameters	Con	Iso	Lir	Iso + Lir
HR 1 (bpm)	243.22 ± 13.09	436.51 ± 7.94 **	245.67 ± 10.75 ††	439.28 ± 14.87 ** ‡‡
HR 2 (bpm)	240.38 ± 4.30	220.88 ± 6.28	237.64 ± 6.05	226.13 ± 11.90
QT 1 (s)	0.173 ± 0.007	0.089 ± 0.005 **	0.167 ± 0.004 ††	0.094 ± 0.004 ** ‡‡
QT 2 (s)	0.170 ± 0.010	0.200 ± 0.017	0.173 ± 0.008	0.188 ± 0.010
QRS 1 (mV)	0.280 ± 0.018	0.405 ± 0.045 *	0.268 ± 0.009 †	0.411 ± 0.042 * ‡
QRS 2 (mV)	0.302 ± 0.033	0.204 ± 0.024 *	0.290 ± 0.019 †	0.210 ± 0.027 * ‡

HR 1—heart rate after the first dose of isoprenaline; HR 2—heart rate before sacrifice; QT 1—QT interval after the first dose of isoprenaline; QT 2—QT interval before sacrifice; QRS 1—QRS peak-to-peak after the first dose of isoprenaline; QRS 2—QRS peak-to-peak before sacrifice; Con—0.9% NaCl s.c. on the first 2 days + 0.9% NaCl s.c. for the next 7 days; Iso—isoprenaline 85 mg/kg s.c. on the first 2 days + 0.9% NaCl s.c. for the next 7 days; Lir—0.9% NaCl s.c. on the first 2 days + liraglutide 1.8 g/kg for the next 7 days; Iso + Lir—isoprenaline 85 mg/kg s.c. on the first 2 days + liraglutide 1.8 g/kg for the next 7 days; * *p* < 0.05 versus Con, ** *p* < 0.001 versus Con, † *p* < 0.05 versus Iso, †† *p* < 0.001 versus Iso, ‡ *p* < 0.05 versus Lir, ‡‡ *p* < 0.001 versus Lir.

## Data Availability

The authors confirm that the data supporting the findings of this study are available within the article.

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
