# Peer review of "Liraglutide Treatment Restores Cardiac Function After Isoprenaline-Induced Myocardial Injury and Prevents Heart Failure in Rats"

_life, 2025, doi:10.3390/life15030443_

Round 1

Reviewer 1 Report

Comments and Suggestions for Authors

This manuscript revealed the usefulness of liraglutide treatment for isoprenaline-induced myocardial injury. I am looking forward to the application of liraglutide treatment to the treatment of heart failure, but I have some questions. Please clarify. 

1. Liraglutide treatment was shown to have cardioprotective effects, such as a decrease in HW/BW ratio and BNP. Could this be due to the reduction in isoprenaline-induced myocardial injury evidenced with TnI? If so, is there a relationship between fibrosis and these heart failure evaluation indices?

2. Similarly, if oxidative stress is one of the pathways by which liraglutide treatment alleviates heart failure, please show the relationship. Since the improvement of blood sugar and lipids by liraglutide treatment reduces oxidative stress. Is this related to the improvement of heart failure? 

3. You also mentioned improvement in liver function in your discussion, but did liraglutide treatment reduce AST/ALT due to improvement of NASH or did the reduction in myocardial injury caused by liraglutide treatment reduce AST/ALT? If it is the former, is it related to improvements in blood sugar and lipids?

4. The changes in the ventricular wall (Fig. 5) seem inconsistent. Do you have any thoughts on this?

Author Response

Dear Reviewer 1,

We appreciate your feedback and suggestions. Our response can be found in the attached Word document.

Kind regards!

Authors

Reviewer 2 Report

Comments and Suggestions for Authors

The ideea of the article started from the fact that many drugs and procedures have been used in order to prevent the development of HF following MI. Isoprenaline is recognized as a model for MI because it provides adequate mimicry and maintains a low mortality rate, reflecting its considerable reliability in translational medicine. In this study, the administration of isoprenaline has been associated with heart enlargement, presented by the rise in H/BW, similar with the observations made in other studies.

Liraglutide is a  GLP-1 receptor agonist (GLP-1 RA) that is commonly used in the treatment of diabetes. The use of GLP-1 RAs is supported by their  beneficial effects on glucose metabolism and the functioning of the pancreas. GLP 1 RA  not only enhance the survival of pancreatic cells but also decrease glycemic levels. The therapeutic effects of GLP-1 RAs are being studied in a wide range of diseases and conditions associated with diabetes. There is evidence that GLP-1 RAs can also improve neurological impairments, including cognitive decline and Alzheimer's disease, dermatological conditions such as psoriasis  and renal diseases. Recent studies have revealed that liraglutide exhibits beneficial effects, including its role in reducing inflammation  and oxidative stress especially at the myocardial level.

The study have established that pretreatment with liraglutide has significant cardioprotective effects in acute MI. Based on the previous studies, this study aimed to evaluate the potential role of liraglutide in the treatment of MI especially in the prevention of heart failure.

The ideea of the study is a very interesting one, the design is clear, well conceived and also very complex.

But in the introduction there are some confusions regarding the definition of myocardial infarction and the difference between myocardial injury and myocardial necrosis. Troponins are elevated in both myocardial injury without infarction and also in myocardial infarction.

I have inserted some of my comments directly into the text.

Also, another source of confusions is related to LDH. Enzymatically active lactate dehydrogenase is consisting of four subunits (tetramer). The two most common subunits are the LDH-M and LDH-H peptides, named for their discovery in muscle and heart tissue, and encoded by the LDHA and LDHB genes, respectively. These two subunits can form five possible tetramers (isoenzymes): LDH-1 (4H), LDH-5 (4M), and the three mixed tetramers (LDH-2/3H1M, LDH-3/2H2M, LDH-4/1H3M). These five isoforms are enzymatically similar but show different tissue distribution.

LDH-1 (4H)—in the heart and in RBC (red blood cells), as well as the brain

LDH-2 (3H1M)—in the reticuloendothelial system

LDH-3 (2H2M)—in the lungs

LDH-4 (1H3M)—in the kidneys, placenta, and pancreas

LDH-5 (4M)—in the liver and striated muscle, also present in the brain.

To which isoenzyme does the study refer to?

References are good.

Author Response

Dear Reviewer 2,

Thank you for taking the time to provide your valuable feedback and suggestions. We truly appreciate your insights. In response to your comments, we have compiled a detailed reply, which you can find in the attached Word document (and revised manuscript).

Kind regards,

Authors

Round 2

Reviewer 1 Report

Comments and Suggestions for Authors

The revised manuscript is much improved. I pay my respects for the authors have made a sincere effort to address concerns. Let me just confirm one thing. 

1. I think the data that can be used as markers of myocardial damage are AST and LDH, excluding ALT. On the other hand, if left heart failure, as indicated by an increase in HW/BW ratio, is progressing to right heart failure and liver damage is emerging, AST and ALT are likely to rise in parallel. However, in the latter case, evidence of right heart failure (lung weight / body weight ratio, etc.) must be provided. 

Author Response

Dear Reviewer,

Thank you very much for your constructive feedback, which we value greatly.  We appreciate your time and effort in reviewing this manuscript. In the accompanying document, you will find a detailed response to your suggestion.

Kind regards!

Authors
